# A Multiomics Evaluation of the Countermeasure Influence of 4-Week Cranberry Beverage Supplementation on Exercise-Induced Changes in Innate Immunity

**DOI:** 10.3390/nu16193250

**Published:** 2024-09-26

**Authors:** David C. Nieman, Camila A. Sakaguchi, James C. Williams, Jongmin Woo, Ashraf M. Omar, Fayaj A. Mulani, Qibin Zhang, Wimal Pathmasiri, Blake R. Rushing, Susan McRitchie, Susan J. Sumner, Jackie Lawson, Kevin C. Lambirth

**Affiliations:** 1Human Performance Laboratory, Appalachian State University, North Carolina Research Campus (NCRC), Kannapolis, NC 28081, USA; olsonca1@appstate.edu (C.A.S.); williamsjc12@appstate.edu (J.C.W.); 2UNCG Center for Translational Biomedical Research, University of North Carolina at Greensboro, North Carolina Research Campus (NCRC), Kannapolis, NC 28081, USA; j_woo@uncg.edu (J.W.); amalsagheer@uncg.edu (A.M.O.); famulani@uncg.edu (F.A.M.); q_zhang2@uncg.edu (Q.Z.); 3Department of Nutrition, University of North Carolina at Chapel Hill, Chapel Hill, NC 27599, USA; wimal_pathmasiri@unc.edu (W.P.); blake_rushing@unc.edu (B.R.R.); susan_mcritchie@unc.edu (S.M.); susan_sumner@unc.edu (S.J.S.); 4Nutrition Research Institute, University of North Carolina at Chapel Hill, North Carolina Research Campus (NCRC), Kannapolis, NC 28081, USA; 5College of Computing and Informatics, University of North Carolina at Charlotte, North Carolina Research Campus (NCRC), Kannapolis, NC 28081, USA; jlawso29@charlotte.edu (J.L.); kclambirth@charlotte.edu (K.C.L.)

**Keywords:** cranberry, exercise, proteomics, oxylipins, gut microbiome, metabolomics, immunity

## Abstract

Objectives: This study examined the effect of a 4-week unsweetened cranberry beverage (CRAN) (317 mg polyphenols) versus placebo beverage (PLAC) ingestion (240 mL/day) on moderating exercise-induced changes in innate immunity. Methods: Participants included 25 male and female non-elite cyclists. A randomized, placebo-controlled, double-blind crossover design was used with two 4-week supplementation periods and a 2-week washout period. Supplementation periods were followed by an intensive 2.25 h cycling bout. Six blood samples were collected before and after supplementation (in an overnight fasted state) and at 0 h, 1.5 h, 3 h, and 24 h post-exercise. Stool and urine samples were collected pre- and post-supplementation. Outcome measures included serum creatine kinase, myoglobin, and cortisol, complete blood counts, plasma untargeted proteomics, plasma-targeted oxylipins, untargeted urine metabolomics, and stool microbiome composition via whole genome shotgun (WGS) sequencing. Results: Urine CRAN-linked metabolites increased significantly after supplementation, but no trial differences in alpha or beta microbiota diversity were found in the stool samples. The 2.25 h cycling bout caused significant increases in plasma arachidonic acid (ARA) and 53 oxylipins (FDR q-value < 0.05). The patterns of increase for ARA, four oxylipins generated from ARA-cytochrome P-450 (CYP) (5,6-, 8,9-, 11,12-, and 14,15-diHETrEs), two oxylipins from linoleic acid (LA) and CYP (9,10-DiHOME, 12,13-DiHOME), and two oxylipins generated from LA and lipoxygenase (LOX) (9-HODE, 13-HODE) were slightly but significantly higher for the CRAN versus PLAC trial (all interaction effects, *p* < 0.05). The untargeted proteomics analysis showed that two protein clusters differed significantly between the CRAN and PLAC trials, with CRAN-related elevations in proteins related to innate immune activation and reduced levels of proteins related to the regulation of the complement cascade, platelet activation, and binding and uptake of ligands by scavenger receptors. No trial differences were found for cortisol and muscle damage biomarkers. Conclusions: CRAN versus PLAC juice resulted in a significant increase in CRAN-related metabolites but no differences in the gut microbiome. CRAN supplementation was associated with a transient and modest but significant post-exercise elevation in selected oxylipins and proteins associated with the innate immune system.

## 1. Introduction

Prolonged and intensive exercise bouts result in transient but significant physiological stress, perturbations in immune function, and increased inflammation. Nutrition-based strategies, including fruit ingestion, can moderate these exercise-induced changes and improve the rate of return to metabolic homeostasis during recovery [1]. Cranberries have a unique nutrient and polyphenol profile, including a high concentration of A-type proanthocyanidins (PACs) that are not found in most other fruits. Other prevalent cranberry polyphenols include phenolic acids and anthocyanins [2]. Cell culture studies support the fact that cranberry polyphenols exert anti-inflammatory, antioxidant, and immune regulatory effects [2,3,4]. The small intestinal absorption of cranberry polyphenols in humans, however, is limited, and most move into the large intestine, where smaller metabolites are generated from bacterial enzyme activity [5]. These metabolites are released into the circulation, where they may exert bioactive and anti-inflammatory effects before being cleared by the kidneys, but human trial evidence of this is limited and inconsistent [2,3,4,5,6,7,8]. An increased intake of cranberry polyphenols may alter the composition of the gut microbiome, but this assumption has not yet been firmly established in randomized, placebo-controlled human clinical trials [5,9,10,11].

Only a few human-based studies have evaluated the potential influence of cranberry ingestion on performance or post-exercise inflammation and oxidative stress, and results have been variable in part due to weaknesses in study designs and limited outcome measurements [12,13,14,15]. A multiomics, human system biology approach is being increasingly used in sports nutrition-based investigations to better understand the complex interactions that occur during physiological stress [1,16]. The use of proteomics and metabolomics with oxylipin analysis by our research group and others has supported the post-exercise anti-inflammatory and metabolic benefits of foods rich in polyphenols or carbohydrates, including blueberries, almonds, astaxanthin, and bananas [17,18,19,20,21,22,23].

This study examined the effect of the 4-week ingestion of 240 mL per day of a cranberry beverage or placebo beverage in palliating innate immune protein changes, inflammation, and other indicators of physiological stress following 2.25 h of intensive cycling. This study employed a randomized crossover design with double-blinded methods. A multiomics approach was utilized that included plasma proteomics, urine metabolomics, a targeted plasma oxylipins analysis, and the whole genome shotgun (WGS) sequencing of stool samples to profile the gut microbiome. The cranberry beverage dosing regimen was based on prior fruit-based studies that supported a 2 to 4-week supplementation period to allow the gut microbiome and peripheral tissues time to adapt to the increased intake [1,16]. We hypothesized that the 4-week intake of an unsweetened polyphenol-rich cranberry beverage would alter the gut microbiome, increase circulating levels of cranberry gut-derived metabolites, and mitigate proteomic and oxylipin inflammation and innate immune system biomarkers following exercise stress induced from 2.25 h of intensive cycling.

## 2. Materials and Methods

### 2.1. Study Participants

Male and female cyclists were invited to take part in this study if they met the inclusion criteria, including being 18 to 55 years of age, capable of cycling 2.25 h in a laboratory setting at a 70% maximal oxygen consumption rate (VO_2max_), and a willingness during the 10-week study period to avoid supplements, herbs, and medications with the potential to influence inflammation, such as non-steroidal anti-inflammatory drugs (NSAIDs). Participants also agreed to limit their intake of cranberries during the study other than what was provided. During the 3-day period prior to the 2.25 h cycling session, subjects agreed to taper exercise training and ingest a moderate carbohydrate diet using a food list restricting high-fat foods and visible fats.

A total of 56 participants were assessed for eligibility, and 29 were entered into the study, with 25 completing all aspects of the protocol (Figure 1). Using data from a study with a similar research design, the study participant number provided more than 80% power to detect a difference in pro-inflammatory oxylipins with an effect size of 1.06 at alpha 0.05 using two-sample *t*-tests [20]. Participants voluntarily signed the informed consent form, and procedures were approved by the university’s Institutional Review Board (IRB 22-0114). Trial Registration: ClinicalTrials.gov, U.S. National Institutes of Health, identifier: NCT05407701.

### 2.2. Study Design

This study employed a randomized, placebo-controlled, double-blind, crossover design with two 4-week supplementation periods and a 2-week washout period. Study participants were required to engage in seven sessions at the Appalachian State University Human Performance Laboratory (HPL) at the North Carolina Research Campus, Kannapolis, NC. In random order, the male and female cyclists supplemented their diets with 240 mL/d of cranberry or a 240 mL/d placebo beverage for 4 weeks, followed by the 2.25 h cycling challenge. Blood, stool, and urine samples were collected pre- and post-4 weeks of supplementation to capture the effect of cranberry beverage intake on the gut microbiome, urine metabolome, and plasma proteome. Additional blood samples were collected immediately post-exercise, and then 1.5 h, 3 h, and 24 h post-exercise to capture the effect of cranberry juice ingestion on the post-exercise plasma proteome and oxylipins.

During the first two lab visits (one week period) prior to the 4-week supplementation period, study participants were given a complete orientation to the study protocol, signed the consent form, reported demographics and training histories using questionnaires, and recorded responses to the delayed onset of muscle soreness (DOMS) 1–10 scale questionnaire [20]. An overnight fasted blood sample was collected. Height and body weight were assessed, with body composition measured using the BodPod system (Cosmed, Rome, Italy). Study participants were tested for maximal aerobic capacity (VO_2max_) during a graded cycling test with the Lode cycle ergometer (Lode B.V., Groningen, The Netherlands) and the Cosmed CPET metabolic cart (Cosmed, Rome, Italy). Urine and stool kits were supplied, and participants provided a 24 h urine sample and a stool sample. Stool samples were collected using the OMNIgene•GUT OMR-200 kit (DNA Genotek Inc., Ontawa, ON, Canada). A 3 d food record was completed with macro- and micro-nutrient intake assessed using the food processor dietary analysis software system (Version 11.11, ESHA Research, Salem, OR, USA). Urine and stool kits and a 3-day food record were given to study participants with instructions to repeat procedures at the end of the 4-week supplementation period prior to reporting to the lab for the 2.25 h cycling bout.

The cranberry and placebo beverages (randomized, counterbalanced order using double-blind procedures) for the first and second 4-week supplementation periods were supplied in coded 240 mL bottles. To facilitate compliance to the supplementation protocol, study participants were contacted via email on a regular basis and returned the coded bottles at the end of the supplementation period. Supplements were ingested daily in split doses, with 120 mL at both the first and last meals of the day (thus, 240 mL/d). The cranberry and placebo beverages were supplied by the sponsor (Ocean Spray Cranberries, Inc., Lakeville-Middleboro, MA, USA) and analyzed for their polyphenol and flavonoid content. The cranberry beverage included 317 ± 19 mg of polyphenols (measured by the Folin–Ciocalteu (F–C) colorimetric assay for total phenolic content), 294 ± 26 mg proanthocyanidins (colorimetric assay using dimethylaminocinnamaldehyde (DMAC)), 41 ± 5 mg anthocyanins (high-performance liquid chromatography with ultraviolet detection (HPLC-UV)), and 9 g of intrinsic sugars per 240 mL serving. Appendix A provides a complete polyphenol and flavonoid summary of the cranberry beverage. The placebo beverage was of similar taste (with citric acid, malic acid, fumaric acid, xanthan gum, and natural flavors), color, and total sugar content. After a 2-week washout period, participants repeated all procedures using the alternative supplement. Participants reported no adverse events from ingesting the supplements over the 4-week period and were 100% compliant with the supplementation regimen. A post-study questionnaire revealed that study participants were unable to correctly determine what beverage they were consuming during the two trials (Χ^2^ = 0.400, *p* = 0.819).

After the 4-week supplementation period, study participants reported to the Human Performance Lab in an overnight fasted state and turned in urine and stool samples, the empty supplement bottles, and the 3 d food record. During the 3-day period prior to the 2.25 h cycling sessions, subjects tapered exercise training and ingested a moderate carbohydrate diet using a food list restricting high-fat foods, visible fats, and cranberries. A fasting blood sample was collected, and a DOMS rating was recorded. Participants ingested 120 mL of the cranberry or placebo beverage and 250 mL of water. After warming up, participants cycled for 2.25 h at approximately 70% VO_2max_ while ingesting water alone (3 mL/kg every 15 min). Participants cycled on their own bicycles fitted to Saris H3 direct drive smart trainers (Madison, WI, USA) with monitoring by the Zwift online training platform (Long Beach, CA, USA) and the Cosmed CPET metabolic cart. Heart rate, cycling speed, cadence, distance, power, breathing rate, ventilation, and oxygen intake were measured after 15 min and then every 30 min during the cycling session. To ensure performance consistency between trials and to focus on the effect of the cranberry beverage on exercise-induced inflammation, performance data from the first trial were used to ensure a similar power and metabolic output during the second trial.

Blood samples and DOMS ratings were collected at 0 h, 1.5 h, 3 h, and 24 h post-exercise. Immediately after the 1.5 h post-exercise blood sample, all subjects consumed 7 kilocalories per kilogram of body weight of a fortified nutrient beverage (Boost, Nestlé S.A., Vevey, Switzerland). After a 2-week washout period, study participants were crossed over to the opposite supplement beverage and then repeated all procedures from the first trial.

### 2.3. Sample Analysis

Serum creatine kinase, myoglobin, and cortisol (from serum separator tubes), and complete blood counts (CBCs) with a white blood cell differential count (EDTA tubes) were analyzed using Labcorp services (Burlington, NC, USA). Plasma aliquots were prepared from EDTA blood collection tubes and stored in a −80 °C freezer for analysis for oxylipins and untargeted proteins. Urine and stool samples were prepared and stored in a −80 °C freezer for analysis for cranberry metabolites and microbiota, respectively.

#### 2.3.1. Plasma Oxylipins and Statistical Procedures

Plasma arachidonic acid (ARA), eicosapentaenoic acid (EPA), docosahexaenoic acid (DHA), and oxylipins were analyzed using a liquid chromatography–multiple reaction monitoring mass spectrometry (LC-MRM-MS) method as fully described elsewhere [24]. Resultant data files were processed with Skyline, and the auto-integrated peaks were inspected manually. Concentrations of each oxylipin were determined from the calibration curves of each analyte, which were constructed by normalizing to the selected deuterated internal standards followed by linear regression with 1/x weighting (Appendix A). The coefficient of variation for the quality control standards was <15%, as reported in the method development paper [24]. A total of 53 out of 75 oxylipins detected increased significantly post-exercise, and these were grouped for statistical analysis. Four other composite variables were calculated including eight oxylipins generated from ARA and cytochrome P-450 (ARA-CYP) (5,6-, 8,9-, 11,12-, and 14,15-dihydroxy-eicosatetraenoic acid (diHETrEs), and 16-, 17, 18-hydroxy-eicosatetraenoic acids (HETEs), and the 20-HETE metabolite 20-carboxy-arachidonic acid (20-coohAA)), a subset of four oxylipins generated from ARA-CYP and soluble epoxide hydrolase (sEH) (5,6-, 8,9-, 11,12-, and 14,15-diHETrEs), two abundant oxylipins generated from linoleic acid (LA) and CYP (9,10-DiHOME, 12,13-DiHOME), and two abundant oxylipins generated from LA and lipoxygenase (LOX) (9-HODE, 13-HODE).

#### 2.3.2. Plasma Proteome

Untargeted proteomics analysis was conducted, as described previously [18,25,26]. Briefly, after sample preparation, 200 ng of peptides from the plasma samples were loaded onto disposable EvoTip trap columns (EV-2003, EvoSep, Odense, Denmark) and separated on an EvoSep One ^TM^ LC system (EV-1000, EvoSep, Odense, Denmark) using a 21 min gradient with 1 µL/min flow rate. Effluents were analyzed on a high-resolution Orbitrap Exploris 240 (Thermo, Waltham, MA, USA) mass spectrometer using the data-independent acquisition (DIA) method. The plasma protein library was generated using the gas-phase fractionation DIA method from peptide samples with and without the depletion of the top 14 high-abundance plasma proteins (14,120 precursors, 960 proteins). The obtained LC-MS/MS dataset was searched for protein identification and quantitation using DIA-NN [27]. Data were normalized by referencing the protein levels of the first time point from the same individual subject to effectively correct for inter-individual variations [28] (Appendix A). The normalized values were statistically analyzed using the ANOVA test with two trials and six time points. To consider the protein as significantly changing between or within effects, the *p*-value was set to less than 0.05. Maximum likelihood-based hierarchical clustering analysis was used to cluster proteins with similar level patterns, and the results were visualized as a heatmap with the averaged value of each time point after normalization by z-score. The list of significantly changed proteins in the enriched clusters were functionally enriched using STRING (Ver.11.5, https://string-db.org/, accessed on 15 August 2023). The top enriched biological processes from STRING analysis were selected to represent the functions of the proteins.

#### 2.3.3. Urine Untargeted Metabolomics Analysis and Statistical Procedures

Untargeted metabolomics analysis procedures have been described in detail elsewhere [29,30,31,32]. Briefly, untargeted metabolomics data of randomized urine samples (interspersed with 10% blanks, quality control study pools (QCSP), and NIST SRM 3672 urine reference material) were acquired in positive mode on a Vanquish UHPLC system coupled with a Q Exactive™ HF-X Hybrid Quadrupole-Orbitrap™ Mass Spectrometer (UHPLC-HRMS; Thermo Fisher Scientific, San Jose, CA, USA). Raw data files for all study samples, QCSP, blank, and NIST reference material runs were uploaded to Progenesis QI (Waters Corporation, Milford, MA, USA) for alignment and peak picking. Data were normalized to a reference QCSP sample using the “normalize to all” function in Progenesis QI [33,34]. Peaks detected by UHPLC-HRMS were identified or annotated using ADAP-KDB software (version 1.8.5) for in-house reference libraries and public databases [35] (Appendix A). The evidence basis for metabolite identifications and annotations was denoted using an ontology system as previously described [32]. As is the case with LC-MS-based platforms, isomers may not always be distinguishable. Names given for each match are based on the names of the reference standards run on the UHPLC-HRMS platform.

The difference in peak intensity for each arm was calculated by subtracting the pre-supplementation peak intensity from the post-supplementation peak intensity. One person was excluded from the analysis due to a missing post-supplementation 24 h urine sample. A linear mixed model was created for each peak using SAS^®^ 9.4 (SAS Institute Inc., 2023, Cary, NC, USA) to determine whether there was a significant effect of cranberry supplementation compared to the placebo. The least square (LS) means were calculated for the difference in peak intensity for the cranberry and placebo trials.

The difference in peak intensity for each trial was also modeled using orthogonal partial least square discriminant analysis [OPLSDA; (SIMCA 17, Sartorius Stedim Data Analytics, AB, Umeå, Sweden)], which is a multivariate method frequently used for analyzing high dimensional collinear data (ref). The variable importance to the projection (VIP) statistic allows for the identification of peaks important to the differentiation of the groups and peaks with a VIP ≥ 1 were considered to be differentiators (citation). The model statistics include R2Y, which is the percentage of variation in the differentiation of the groups explained by the model, and Q2, which is based on a 7-fold cross-validation that provides an assessment of the predictive ability of the model.

#### 2.3.4. Gut Microbiome Analysis and Statistical Procedures

DNA isolation: fecal samples were transferred to a 2 mL tube containing 200 mg of 106/500 μm glass beads (Sigma, St. Louis, MO, USA) and 0.5 mL of the Qiagen PM1 buffer (Valencia, CA, USA). Mechanical lysis was performed for 40 min on a Digital Vortex Mixer. After 5 min of centrifugation, 0.45 mL of supernatants were aspirated and transferred to a new tube containing 0.15 mL of Qiagen IRS solution. The suspension was incubated at 4 °C overnight. After brief centrifugation, the supernatant was aspirated and transferred to deep well plates containing 0.45 mL of the Qiagen binding buffer supplemented with Qiagen ClearMag Beads. DNA was purified using the automated KingFisher™ Flex Purification System and eluted in DNase-free water [36,37,38].

##### Illumina Whole Genome Shotgun (WGS) Sequencing [39]

In total, 5 ng of genomic DNA were processed using the Nextera XT DNA Sample Preparation Kit (Illumina, Inc. San Diego, CA, USA). Target DNA was simultaneously fragmented and tagged using the Nextera Enzyme Mix containing transposome that fragments the input DNA and adds the bridge PCR (bPCR)-compatible adaptors required for binding and clustering in the flow cell. Next, fragmented and tagged DNA was amplified using a limited-cycle PCR program. In this step, index 1(i7) and index 2(i5) were added between the downstream bPCR adaptor and the core sequencing library adaptor, as well as the primer sequences required for cluster formation. The thermal profile for the amplification had an initial extension step at 72 °C for 3 min and an initial denaturing step at 95 °C for 30 s, followed by 15 cycles of denaturing at 95 °C for 10 s, annealing at 55 °C for 30 s, a 30 s extension at 72 °C, and final extension for 5 min at 72 °C. The DNA library was then purified using the Agencourt^®^ AMPure^®^ XP Reagent. Each sample was quantified and normalized before pooling. The DNA library pool was loaded on the Illumina platform reagent cartridge and the Illumina instrument [40]. For validation of the DNA isolation process, a known bacterial community, ZymoBIOMICS Gut Microbiome Standard (Cat# D6331) (Zymo Research, Irvine, CA, USA), and blanks composed of only DNA isolation reagents were included in the DNA extraction process and again in library preparation. In addition to the isolation controls, the library preparation included library blanks composed of library preparation reagents alone.

###### Bioinformatics

Sequencing outputs from the Illumina NextSeq 4000 platform were converted to fastq format and demultiplexed using Illumina Bcl2Fastq 2.20.0 (Illumina, Inc. San Diego, CA, USA). The quality control of the demultiplexed sequencing reads was verified by FastQC (Babraham Institute. Cambridge, UK). Adapters were trimmed using Trim Galore (Babraham Institute. Cambridge, UK). The resulting paired-end reads were submitted to Kraken2 for taxonomic classification [41]. An estimate of taxonomic composition, including the host, was produced from these results using Bracken 2.5 [42]. All reads classified as the host were eliminated, and paired-end reads were joined with vsearch 2.7.0 [43]. Estimates of taxonomic composition, gene family, path abundance, and path coverage were produced from the remaining reads using HUMAnN3 [44]. Alpha diversity, as measured by Evenness, beta diversity, as measured by Bray–Curtis dissimilarity, and ANCOM differential abundance analysis, was performed using QIIME2 [45] (Appendix A).

### 2.4. Additional Statistical Procedures

Data are expressed as the mean ± SE. Except where described, datasets were analyzed using the generalized linear model (GLM) and repeated measures ANOVA module in SPSS (IBM SPSS Statistics, Version 28.0, IBM Corp, Armonk, NY, USA). The statistical model utilized the within-subjects approach: 2 (trials) × 6 (time points) repeated measures of ANOVA and provided time (i.e., the collective effect of the cycling exercise bout) and interaction effects (i.e., whether the data pattern over time differed between trials). If the interaction effect was significant (*p* ≤ 0.05), then post hoc analyses were conducted using paired t-tests comparing timepoint contrasts between trials. An alpha level of *p* ≤ 0.01 was used after Bonferroni correction for 5 multiple tests. The positive false discovery rate (FDR or “q-value”) was calculated for the multiple testing correction of the plasma oxylipin and plasma proteomics data. Violin plots were also generated in RStudio (version 2024.04.2) using the packages ggplot2 (version 3.5.1), ggdist (version 3.3.2), and RColorBrewer (version 1.1-3).

## 3. Results

A total of 25 study participants (*n* = 17 males, *n* = 8 females) completed all study procedures (Table 1). Age and the percentage of body fat were comparable between sex groups, with higher aerobic (VO_2max_) and ventilation capacity for the male versus female cyclists. As designed to ensure similar physiological stress states, performance data for the cranberry and placebo were comparable in average watts of power, heart rates, oxygen consumption rates, the total distance cycled, and speed (Table 2). The average percentage of maximum heart rate and VO_2max_ during both trials did not differ significantly between sex groups. Additionally, the pattern of change over time did not differ between the male and female cyclists for two key outcome measurements (total plasma oxylipins, supplement × time × sex interaction effect, *p*-value = 0.464; gut microbiome alpha diversity, supplement × time × sex interaction effect, *p*-value = 0.573). Thus, outcome measures are presented for all study participants combined.

Three-day food records were collected at the beginning and end of each 4-week supplementation period. No differences in macronutrient intake were found within or between trials. Nutrient data from the four 3-day food records were averaged for the 25 cyclists. Energy intake averaged 2144 ± 112 kcal/day (8.97 ± 0.47 MJ/day), and carbohydrates, protein, fats, and alcohol represented 41.1 ± 1.5, 19.5 ± 0.9, 36.9 ± 1.0, and 3.7 ± 0.7%, respectively, of the total energy. The total flavonoid intake of the background diet averaged 71.6 ± 12.6 mg/day, with flavan-3-ol, anthocyanin, and flavanol intake averaging 30.4 ± 8.2, 20.1 ± 4.9, and 13.3 ± 1.2 mg/day.

Significant post-exercise increases were measured for the delayed onset of muscle soreness (DOMS), muscle damage biomarkers serum myoglobin and creatine kinase, white blood cell counts and the neutrophil/lymphocyte ratio, and serum cortisol in both the cranberry and placebo trials, with no differences in the pattern of change over time (all interaction effects, *p* > 0.20.

OPLSDA analysis of the untargeted metabolomics data (12,818 peaks) from urine samples collected 24 h before and after each supplementation period indicated a strong cranberry versus placebo trial separation (R2Y = 0.98) and a moderate indication that the data were reproducible (Q2 = 0.416, Figure 2a) even with the small sample size used in this study. The distribution of the difference in peak intensity (post-supplementation minus pre-supplementation or T2-T1) for six metabolites is shown in Figure 2b. These metabolites had a variable importance in the projection (VIP) of more than 2.0, an ontology level that included a retention time match, an FDR *p*-value < 0.05, and literature confirmation as a cranberry-related metabolite. These included significant increases in hippuric acid, 4-hydroxybenzaldehyde, 4-hydroxycinnamic acid, 4-coumarate, isoferulic acid, and caffeic acid (other potential matches for these peaks are provided in Appendix A). Appendix A provides the VIP and *p*-values for the top 90 metabolites that matched the in-house library at an ontology level of OL_1 or OL2a.

Plasma arachidonic acid (ARA), docosahexaenoic acid (DHA), and eicosapentaenoic acid (EPA) increased significantly post-exercise (time effects, *p* < 0.001) (Table 3). The pattern of change for ARA but not DHA or EPA differed between the cranberry and placebo trials, with a higher increase in ARA post-exercise for the cranberry trial (interaction effect, *p* = 0.027). A total of 75 oxylipins were detected and quantified in the study samples. Of these, 53 oxylipins increased significantly post-exercise (FDR q-value < 0.05) (Appendix A). These 53 oxylipins were summed for a composite variable, and analysis showed significant post-exercise increases without trial differences between the cranberry and placebo trials (interaction effect, *p* = 0.189) (Table 3). Four other composite variables were calculated, including eight oxylipins generated from ARA and cytochrome P-450 (ARA-CYP) (5,6-, 8,9-, 11,12-, and 14,15-dihydroxy-eicosatetraenoic acid (diHETrEs), 16-, 17, 18-hydroxy-eicosatetraenoic acids (HETEs), the 20-HETE metabolite 20-carboxy-arachidonic acid (20-coohAA)), a subset of four oxylipins generated from ARA-CYP and soluble epoxide hydrolase (sEH) (5,6-, 8,9-, 11,12-, and 14,15-diHETrEs), two abundant oxylipins generated from linoleic acid (LA) and CYP (9,10-DiHOME, 12,13-DiHOME), and two abundant oxylipins generated from LA and lipoxygenase (LOX) (9-HODE, 13-HODE) (Table 3 and Figure 3). Significant time effects were shown for each of these composite oxylipin variables, with significant interaction effects and higher levels post-exercise for the cranberry versus placebo trial for the diHETrEs (*p* = 0.003) (Figure 3), DiHOMEs (*p* = 0.022), and the HODEs (*p* = 0.008) (Table 3).

The nanoLC-MS/MS untargeted proteomics analysis identified 595 plasma proteins. Of these, 467 proteins had no missing data across all six timepoints. The data were normalized to the first time point; the Z-score was transformed and analyzed using hierarchical clustering. Two clusters were identified that differed significantly between the cranberry and placebo trials (Figure 4 and Figure 5). Proteins enriched in reactome pathway analysis were visualized with their STRING DB interactions if the edge had a combined score greater than 0.35. The size and color of the nodes in Figure 5a,b indicate the *p*-value (overall supplementary effect) of the two-way ANOVA tests. Cluster A proteins were elevated in the cranberry versus placebo trial and were most strongly related to the innate immune system (FDR q-value = 0.00012), hemostasis (q = 0.0027), neutrophil degranulation (q = 0.0028), and complement cascade (q = 0.0030) reactome pathways. Cluster B proteins were lower in the cranberry versus placebo trial and were most strongly related to platelet degranulation (q = 1.21 × 10^−7^), the regulation of insulin-like growth factor (IGF) transport (q = 9.47 × 10^−7^), platelet activation, signaling and aggregation (q = 9.47 × 10^−7^), platelet activation, signaling, and aggregation (q = 9.47 × 10^−7^), the binding and uptake of ligands by scavenger receptors *q = 4.82 × 10^−5^), neutrophil degranulation (q = 0.0028), and the regulation of the complement cascade (q = 7.53 × 10^−5^) reactome pathways. If a protein had four or more interactions and was statistically significant when comparing the cranberry and placebo trials, it was chosen as a key protein in the pathway [Table 4 (cluster A) and Table 5 (cluster B)].

Whole genome shotgun (WGS) sequencing was conducted on the stool samples collected before and after 4-weeks of supplementation with cranberry or placebo beverages. The sequencing output was converted, and alpha diversity was calculated by evenness and beta diversity for Bray–Curtis dissimilarity. ANCOM differential abundance analysis was performed using QIIME2. A total of 5719 taxa were identified, with 339 that were present in more than 20% of stool samples used in the analysis. Figure 6 depicts the box whisker plots for the pre- and post-4 week supplementation of stool sample microbiome alpha diversity (observed richness or the number of taxa) with cranberry and placebo beverages. No significant time or interaction effects were found (time effect *p* = 0.650, interaction effect *p* = 0.302). PERMANOVA analysis for beta diversity showed no differences in the microbiome profiles between the samples (F = 0.350, *p* = 0.993). No genus or species differences were found after 4-weeks of cranberry consumption compared to placebo beverage intake.

## 4. Discussion

An intake of 240 mL/day of unsweetened cranberry beverage compared to the placebo caused a robust urine cranberry metabolite signature without changes in gut microbiome alpha and beta diversity. As expected, and in accordance with previous reports from our research group, the 2.25 h cycling bout triggered significant increases in plasma oxylipins and proteins linked to innate immune system activation [18,20,21,25]. Cranberry versus placebo beverage intake was associated with small but significantly greater post-exercise elevations in pro-inflammatory oxylipins and a cluster of proteins from the innate immune system, hemostasis, neutrophil degranulation, and complementary cascade molecular pathways. Cranberry beverage intake was also related to lower post-exercise plasma levels of the proteins involved in the regulation of the complement cascade, the binding and uptake by scavenger receptors, platelet degranulation, and IGF regulation. No trial differences were found for serum cortisol, muscle soreness, muscle damage biomarkers, or white blood cell counts.

Few studies have investigated the countermeasure effect of cranberry ingestion on post-exercise physiological measures, and the results have been inconsistent due to disparate research designs, dosing regimens, and limited outcome measures [12,13,14,15]. This study employed a multiomics approach with proteomics, the targeted measurement of oxylipins, urine metabolomics, and stool WGS sequencing to broaden scientific understanding regarding the effect of the 4-week intake of the cranberry beverage on exercise inflammation resolution. Our findings differ markedly from what we and others have reported following blueberry ingestion, which is a fruit that is high in B-type PACs, anthocyanins, and sugars [20,46]. In one study using a parallel group trial with 59 cyclists, 2 weeks of ingestion of 26 g/d blueberries (one cup equivalent per day, 1059 mg total polyphenols) increased plasma levels of 24 gut-derived phenolics and lowered ARA-CYP pro-inflammatory oxylipins following a 75 km cycling bout [20]. A modest negative relationship between 1.5 h post-exercise plasma levels of blueberry metabolites and ARA-CYP oxylipins was also shown. CYP enzymes may be influenced by upward shifts in circulating gut-derived blueberry metabolites [20]. The inhibitory influence of different types of flavonoids on CYP enzymes is an emerging area of scientific endeavor, and little is known about the in vivo effect of gut-derived phenolics [47].

The polyphenol profile is distinctly different between cranberries and blueberries, with cranberries being one of the few fruits containing A-type PACs. PACs are common dietary polyphenols and consist of oligomers and polymers of flavan-3-ol monomeric subunits, including (+)-catechin, (−)-epicatechin, (+)-gallocatechin, and epigallocatechin. The majority of ingested PACs reach the colon, where they are metabolized by the gut microbiota into smaller metabolites including benzaldehydes, benzene derivatives, and phenolic acids, as shown in this study [2,10]. Cell culture and animal studies indicate that both A- and B-type PACs exert anti-inflammatory effects, but randomized human clinical trials have not consistently supported these findings [2,3,4,5,6,7,8,48]. A-type PACs in cranberries have a more complex linkage than B-type PACs, and their biosynthesis and biological activities are less understood [49]. A-type PACs exert antibacterial and antiviral effects by inhibiting bacterial adhesion and virus replication, reduce the risk for urinary tract infections, have positive effects on the integrity and functionality of the urothelium and intestinal epithelium, reduce inflammation within the intestine, and may lower platelet hyperaggregability [49,50,51,52]. The underlying mechanisms explaining the modest but significant increase in post-exercise ARA-CYP, LA-CYP, and LA-LOX oxylipins with cranberry intake warrants further investigation. Another consideration is the dosing regimen chosen for the current study. The intake of the cranberry beverage was 240 mL/d and supplied 317 mg/d polyphenols, which is a dose well below what is supplied with a 26 g/d of freeze-dried blueberry powder [12,17,20]. On the day of the exercise challenge bouts, half doses of the cranberry beverage were consumed immediately prior to exercise compared to full doses of the blueberry supplement 1.5 h after exercise. Cranberry supplements come in several forms, including freeze-dried cranberry powder, dried whole cranberries, cranberry juice cocktails, or capsules containing concentrated extracts. The PAC and polyphenol levels in these different types of cranberry supplements vary widely and may have different physiological effects.

As shown in the current study, prolonged and intensive exercise has a strong transient effect in altering the plasma proteome with perturbations in proteins related to humoral immunity, and inflammation, and innate immune system activation [18,25,53]. In two recent studies, we showed that both astaxanthin and beet supplements partially countered extensive alterations in immune-related proteins following 2.25 h of intensive running or cycling [18,25]. The strongest effect of 4-week astaxanthin supplementation was in mitigating post-exercise decreases in plasma immunoglobulins [18]. Plasma levels of IgM decreased significantly after running 2.25 h but recovered after the 24 h post-exercise recovery period in the astaxanthin but not the placebo trial. The ingestion of a mixed beet supplement during a 2-week period countered increases in proteins related to complement activation, the acute phase response, and immune cell adhesion, migration, and differentiation after 2.25 h of intensive cycling [25]. The effects of diet-based supplements on the plasma proteome within an exercise stress context using untargeted proteomics is a nascent science [1,16]. In the current study, cranberry intake was linked to an upregulation of innate immune system proteins, contrary to what was measured with beet supplementation.

The metabolic significance of these findings in terms of recovery from physiological stress remains to be determined. The use of diet-based supplements to temper exercise-induced inflammation and physiological stress has been challenged as interfering with signaling pathways for adaptation and recovery [20]. Our research group has focused on moderating but not totally countering the undue inflammation and innate immune system perturbations linked to physiologically stressful levels of intensive exercise workloads through fruit-based increases in carbohydrate and polyphenol intake [20].

The health and anti-inflammatory effects of cranberry intake appear to depend on the gut microbiome where metabolites are generated, and potential alterations in its composition occur [5]. Several randomized human clinical trials have been conducted to determine the influence of cranberry intake on gut microbiome composition and alpha and beta diversity with varying results [11,54,55,56,57]. One uncontrolled study with 28 human subjects showed that just 4 days of cranberry extract supplementation (109 mg/d polyphenols and 125 mg oligosaccharides) encouraged the growth of bifidobacteria and butyrate-producing bacteria such as Clostridium and Anaerobutyricum [9]. Further randomized controlled studies are needed. A study with mice showed that 14 days of intake of a high-dose cranberry juice extract was associated with an increase in Akkermansia muciniphila and Clostridium hiranonis [55]. In general, the intake of the cranberry beverage (240 mL/d) used in this 4-week study did not appear to alter the gut microbiome’s composition or diversity.

## 5. Conclusions

This randomized crossover study used an intense 2.25 h cycling bout to induce inflammation and innate immune system perturbations in 25 cyclists who consumed 240 mL/d of unsweetened cranberry or placebo beverages for 4 weeks. Unsweetened cranberry beverage intake was associated with a transient and modest post-exercise elevation in pro-inflammatory oxylipins and proteins associated with the innate immune system but without effects on serum cortisol and muscle damage biomarkers. Additionally, cranberry beverage intake did not alter the gut microbiome composition or diversity. Underlying mechanisms linked to these findings remain to be determined but may be associated with the dosing regimen, the macronutrient and polyphenol composition, and the presence of other bioactives within the cranberry beverage used in this study. This study and others with a similar research design demonstrate that the polyphenol profiles of different types of fruits vary widely and, depending on the dosing regimen, may have contrasting effects on metabolic recovery from stressful levels of exercise [12,17,20,22,23].

## Figures and Tables

**Figure 1 nutrients-16-03250-f001:**
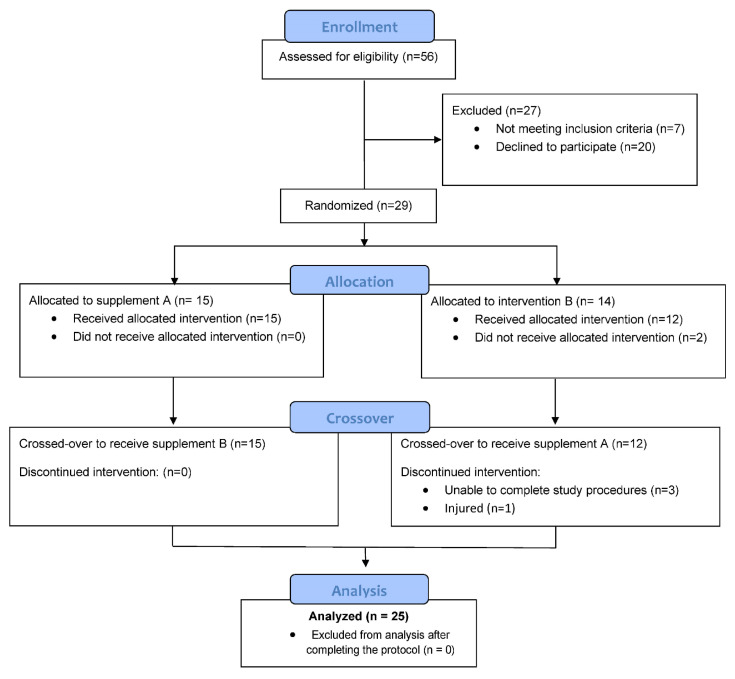
Study participant flow diagram.

**Figure 2 nutrients-16-03250-f002:**
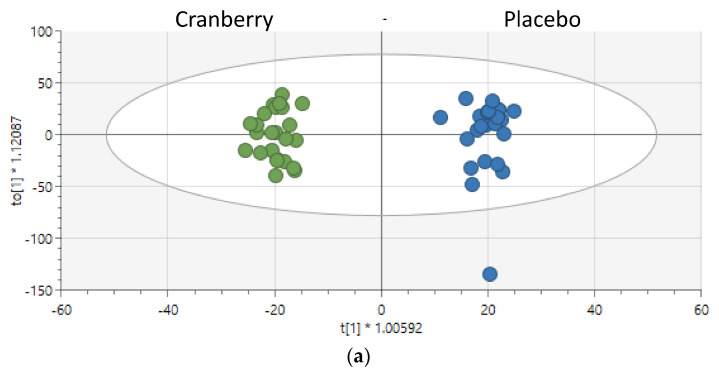
(**a**) OPLSDA analysis of 24 h urine samples collected before and after the 4-week supplementation period. R2X = 0.142, R2Y = 0.98, Q2 = 0.416. Data represent metabolite changes when comparing post- and pre-supplementation urine samples. (**b**) Urine metabolites that were higher in the cranberry trial supported the trial separation data from (**a**). Metabolites were selected that met these criteria: VIP > 2.0, FDR *p* < 0.05, high ontology annotation, and literature confirmation as a cranberry-related metabolite. Box plots for the six metabolites show the distribution of the 4-week change in peak intensity for the cranberry and placebo trials.

**Figure 3 nutrients-16-03250-f003:**
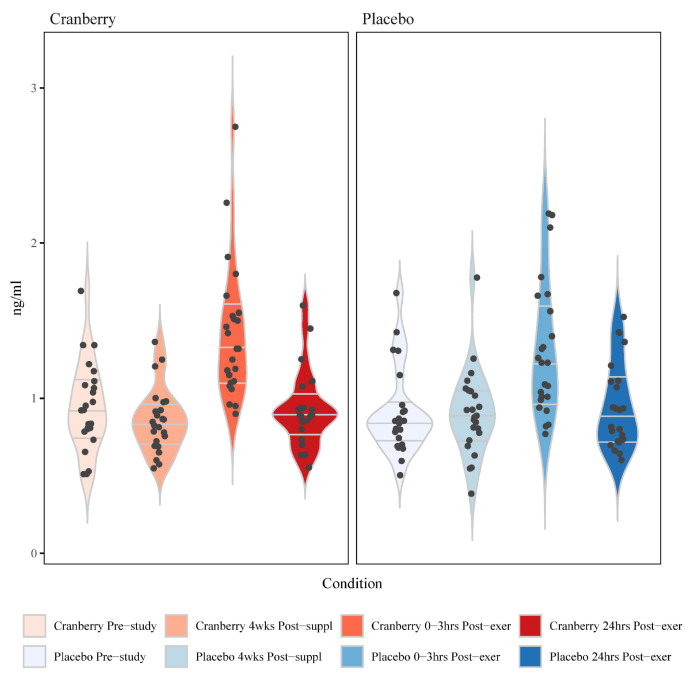
Violin plots for the change in the sum of 4 plasma DiHETrEs between the cranberry and placebo trials (time effect *p* < 0.001, interaction effect *p* = 0.003). The horizontal lines represent the median and interquartile range. Wider sections of the violin plot represent a higher probability that study participants will take on the given value.

**Figure 4 nutrients-16-03250-f004:**
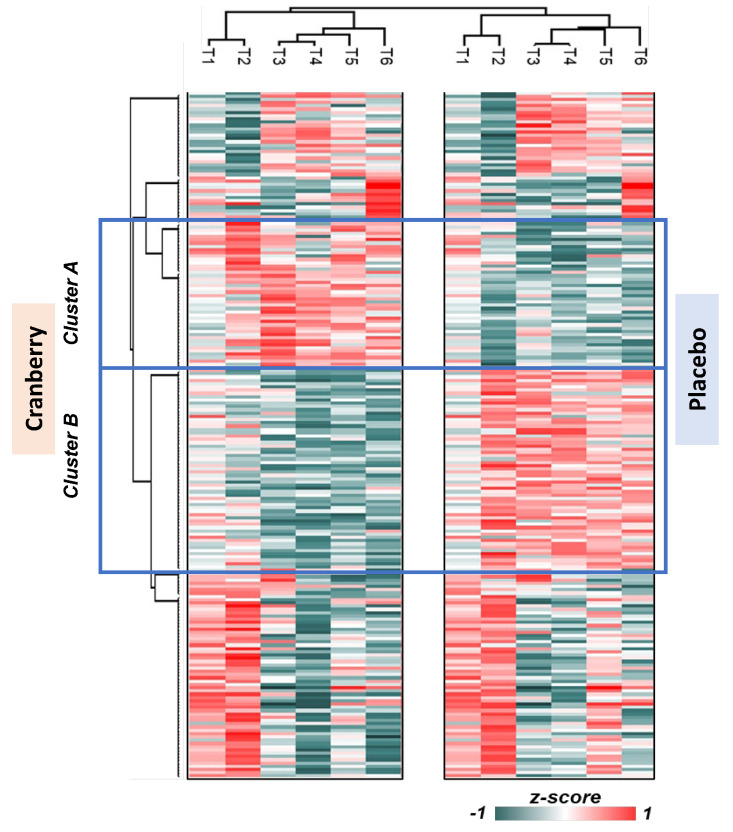
Untargeted proteomics heat map for the cranberry and placebo trials across six timepoints (T1 = pre-supplementation, T2 = post-4 weeks supplementation, T3 = 0 h post-exercise, T4 = 1.5 h post-exercise, T5 = 3 h post-exercise, and T6 = 24 h post-exercise).

**Figure 5 nutrients-16-03250-f005:**
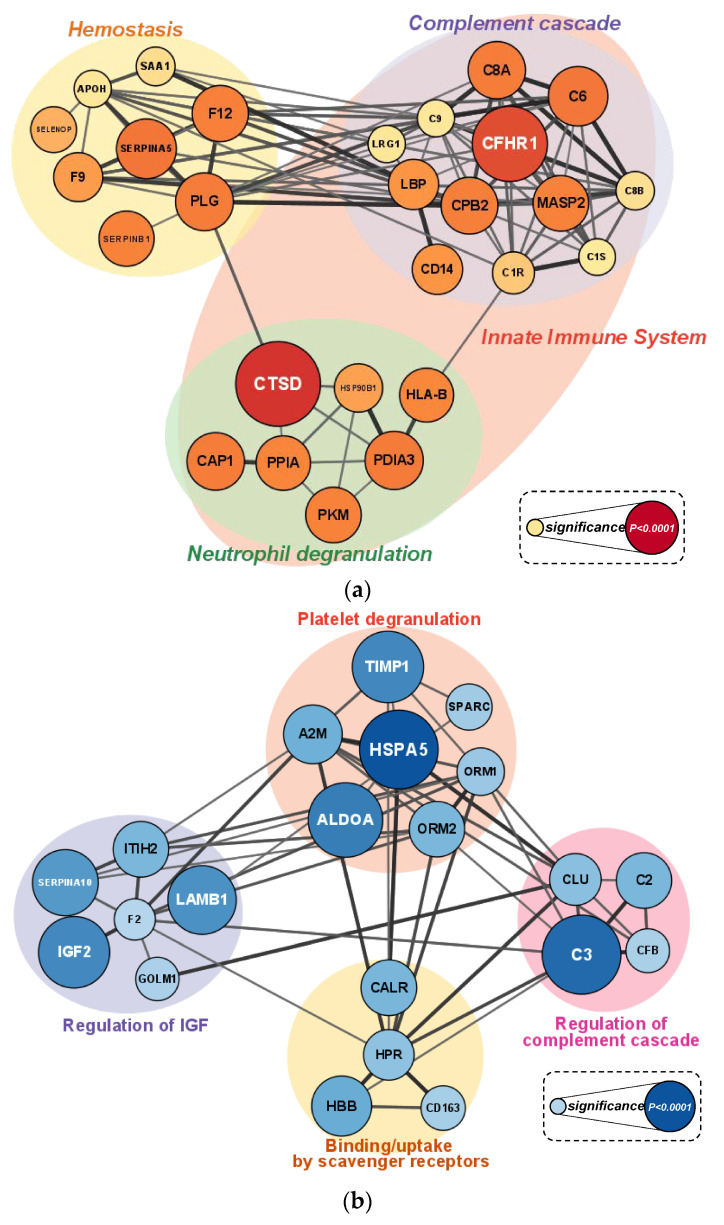
(**a**) Cluster A (elevated in the cranberry trial) visualizes the protein–protein interaction (PPI) network for proteins enriched in reactome pathway analysis. The size and color of the nodes indicate the *p*-value from the two-way ANOVA analysis. Proteins with four or more interactions and statistical significance were chosen as key proteins in the pathway. (**b**) Cluster B (lower in the cranberry trial) visualizes the protein–protein interaction (PPI) network for proteins enriched in reactome pathway analysis. The size and color of the nodes indicate the *p*-value from the two-way ANOVA analysis. Proteins with four or more interactions and statistical significance were chosen as key proteins in the pathway. See Table 4 and Table 5 for an explanation of the abbreviations.

**Figure 6 nutrients-16-03250-f006:**
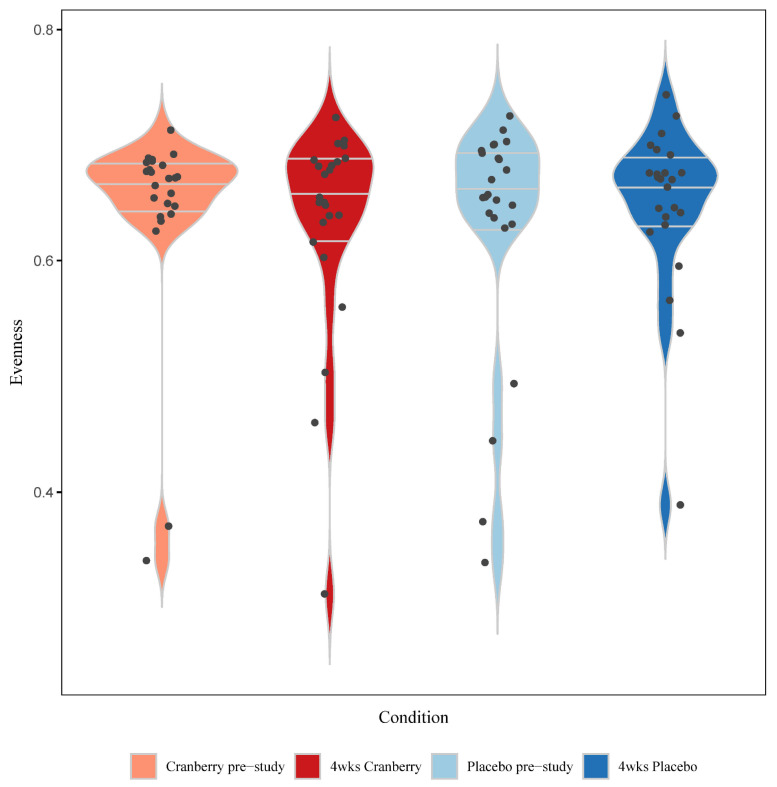
Violin plots for the pre- and post-4 week supplementation of stool sample microbiome alpha diversity (observed richness or the number of taxa) with cranberry and placebo beverages. Time effect *p* = 0.650; interaction effect *p* = 0.302. The horizontal lines represent the median and interquartile range. Wider sections of the violin plot represent a higher probability that study participants would take on the given value.

**Table 1 nutrients-16-03250-t001:** Subject characteristics (*n* = 25) for male (M) (*n* = 17) and female (F) (*n* = 8) cyclists. * Sex contrast, *p*-value ≤ 0.05.

	Sex	Mean ± SE
Age (years)	M	43.2 ± 2.2
	F	41.8 ± 2.9
Weight (kg)	M	79.8 ± 2.5 *
	F	61.4 ± 4.0
Height (cm)	M	178 ± 0.9 *
	F	165 ± 2.4
Body Mass Index (BMI) (kg/m^2^)	M	25.2 ± 0.7 *
	F	22.4 ± 1.1
Body fat (%)	M	22.1 ± 1.7
	F	26.0 ± 3.6
VO_2max_ (mL.kg^−1^ min^−1^)	M	46.2 ± 2.1 *
	F	37.4 ± 2.1
Max watts	M	269 ± 7.3 *
	F	216 ± 22.1
Max heart rate (beats/min)	M	172 ± 2.1
	F	165 ± 3.4
Max ventilation (L/min)	M	136 ± 5.5 *
	F	83.1 ± 4.7
Max respiratory rate (breaths/min)	M	46.6 ± 2.2
	F	45.0 ± 1.6

**Table 2 nutrients-16-03250-t002:** Cycling performance outcomes for the cranberry and placebo trials (*n* = 25 male and female cyclists combined) (mean ± SE). No significant trial differences for an average of 2.25 h of cycling performance measurements were found, as designed. For both trials, lab temperature average was 21.1 ± 0.4 °C and humidity was 45.8 ± 1.0%.

Performance Measurement		Mean ± SE
Cycling power (watts, % maximum)	Cranberry	159 ± 7.2 (62.8 ± 1.7% max)
	Placebo	153 ± 6.9 (60.9 ± 2.0% max)
Heart rate (beats/min, % maximum)	Cranberry	133 ± 2.3 (78.5 ± 1.5% max)
	Placebo	134 ± 2.9 (78.7 ± 1.7% max)
Oxygen consumption (VO_2_) (mL.kg^−1^ min^−1^, % maximum)	Cranberry	29.8 ± 1.3 (69.1 ± 1.7% max)
	Placebo	28.6 ± 1.3 (66.6 ± 2.3% max)
Distance cycled (km)	Cranberry	61.2 ± 2.7
	Placebo	61.6 ± 2.0
Speed (km/h)	Cranberry	27.5 ± 0.8
	Placebo	26.5 ± 0.9

**Table 3 nutrients-16-03250-t003:** Trial comparisons (cranberry = CRAN and placebo = PLAC) across six timepoints for plasma arachidonic acid (ARA), docosahexaenoic acid (DHA), eicosapentaenoic acid EPA), and oxylipin subgroups (mean ± SE). *p*-values represent time (first value) and trial × time interaction effects. Changes from pre-supplementation were not significantly different when contrasted between trials.

Variable (ng/mL)	Trial	Pre- Suppl.	4 Weeks Suppl.	0 h Post-Ex	1.5 h Post-Ex	3 h Post-Ex	24 h Post-Ex	*p*-Value
Arachidonic acid (ARA)	CRAN	1122 ± 68.1	1084 ± 65.5	2592 ± 151	2015 ± 74.8	910 ± 49.9	1009 ± 52.5	<0.001;
	PLAC	1050 ± 61.0	1015 ± 56.3	2467 ± 123	1719 ± 72.4	829 ± 40.3	1029 ± 44.3	0.027
Docosahexaenoic acid (DHA)	CRAN	323 ± 40.3	336 ± 41.8	932 ± 81.3	748 ± 65.1	264 ± 32.7	314 ± 42.4	<0.001;
	PLAC	336 ± 45.3	278 ± 25.1	858 ± 87.6	622 ± 56.7	218 ± 19.4	297 ± 29.7	0.183
Eicosapentaenoic acid (EPA)	CRAN	278 ± 41.8	284 ± 51.3	687 ± 90.9	548 ± 77.9	238 ± 34.2	288 ± 63.6	<0.001;
	PLAC	292 ± 64.3	214 ± 20.1	617 ± 72.5	417 ± 46.4	189 ± 17.1	239 ± 27.0	0.104
Oxylipins, total (*n* = 53 oxylipins)	CRAN	46.5 ± 4.3	45.0 ± 2.6	94.3 ± 7.6	76.7 ± 5.8	54.0 ± 3.0	42.3 ± 2.6	<0.001;
	PLAC	47.9 ± 3.2	43.8 ± 2.6	92.9 ± 7.6	65.4 ± 4.2	47.4 ± 2.3	44.0 ± 2.6	0.189
ARA-CYP † (*n* = 8 oxylipins)	CRAN	7.2 ± 0.7	6.6 ± 0.4	12.7 ± 1.2	16.5 ± 1.9	13.6 ± 1.5	6.6 ± 0.4	<0.001;
	PLAC	6.3 ± 0.3	6.4 ± 0.4	12.9 ± 1.9	15.0 ± 2.2	12.3 ± 1.5	7.1 ± 0.5	0.446
LA-CYP DiHOMES (9,10 + 12,13) ††	CRAN	3.2 ± 0.3	4.8 ± 0.5	12.4 ± 1.5	6.6 ± 1.0	5.6 ± 0.4	4.2 ± 0.6	<0.001;
	PLAC	4.7 ± 0.6	3.7 ± 0.4	11.6 ± 1.2	6.2 ± 0.7	5.5 ± 0.4	4.3 ± 0.6	0.022
LA-LOX HODES (9 + 13) †††	CRAN	5.4 ± 0.5	6.6 ± 0.5	17.6 ± 1.7	10.8 ± 1.2	5.3 ± 0.5	6.1 ± 0.6	<0.001;
	PLAC	6.6 ± 0.6	5.5 ± 0.5	17.0 ± 1.5	8.9 ± 0.8	4.8 ± 0.3	6.0 ± 0.7	0.008

† ARA-CYP = 8 oxylipins generated from arachidonic acid and cytochrome P-450 (ARA-CYP) were grouped and these included 5,6-, 8,9-, 11,12-, and 14,15-dihydroxy-eicosatetraenoic acid (diHETrEs), 16-, 17-, 18-hydroxy-eicosatetraenoic acids (HETEs), and the 20-HETE metabolite 20-carboxy-arachidonic acid (20-coohAA). †† DiHOMES = dihydroxy-9Z-octadecenoic acid. LA = linoleic acid. ††† HODES = hydroxy-octadecadienoic acid. LOX = lipoxygenase.

**Table 4 nutrients-16-03250-t004:** The top 10 key proteins associated with cluster A that were higher in the cranberry versus placebo trial.

Protein	Two-Way ANOVA *p* Value Supplement Effect	# of Interactions
Cathepsin D (CTSD): degrades proteins and activates precursors of bioactive proteins in pre-lysosomal compartments.	<0.0001	4
Complement factor H related 1 (CFHR1): involved in regulating innate immune complement reactions.	0.0003	8
Protein C inhibitor, plasminogen activator inhibitor-3 (SERPINA5): serine protease inhibitor that limits protein C.	0.0078	5
Complement component C6 (C6): a complement system protein involved in the membrane attack complex (MAC).	0.0083	11
Protein disulfide isomerase family A member 3 (PDIA3): a chaperone protein that mediates protein folding.	0.0108	5
Plasminogen (PLG): dissolves the fibrin of blood clots; it is a proteolytic factor in a variety of other processes, including inflammation.	0.0117	12
Complement component C8 alpha chain A (C8A): a constituent of MAC that plays a key role in innate and adaptive immune responses.	0.0131	9
Carboxypeptidase B2 (CPB2): it downregulates fibrinolysis.	0.0144	12
Coagulation factor XII (F12): undertakes the initiation of blood coagulation, fibrinolysis, and the generation of bradykinin and angiotensin.	0.0164	8
Mannan-binding lectin serine protease 2 (MASP2): a serum protease involved in complement system activation.	0.0186	9

**Table 5 nutrients-16-03250-t005:** The top 10 key proteins associated with cluster B were lower in the cranberry versus placebo trial.

Protein	Two-Way ANOVA *p* Value Supplement Effect	# of Interactions
Heat shock protein family A (Hsp70) member 5 (HSPA5): HSP70 chaperone involved in the folding and assembly of proteins in the endoplasmic reticulum.	<0.0001	5
Complement component C3 (C3): plays a central role in the activation of the complement system.	<0.0001	11
Metalloproteinase inhibitor 1 (TIMP1): an inhibitor of the matrix metalloproteinases (MMPs), which is a group of peptidases involved in degradation of the extracellular matrix.	0.0001	9
Serpin family A member 10 (SERPINA10): inhibits the activity of coagulation factors Xa and XIa in the presence of protein Z, calcium, and phospholipid.	0.0002	5
Hemoglobin subunit beta (HBB): involved in oxygen transport from the lung to the various peripheral tissues.	0.0009	5
Alpha-2-macroglobulin (A2M): protease inhibitor; inhibits inflammatory cytokines and disrupts inflammatory cascades.	0.0012	14
Calreticulin (CALR): a calcium-binding chaperone that promotes folding, oligomeric assembly, and quality control in the endoplasmic reticulum.	0.0025	5
Orosomucoid 2 (ORM2): an acute-phase reactant.	0.0026	8
Clusterin (CLU): involved in numerous processes, including the regulation of complement activity and the clearance of cellular debris and apoptosis.	0.0067	12
Haptoglobin-related protein (HPR): associated with apolipoprotein L-I (apoL-I)-containing high-density lipoprotein (HDL).	0.0087	12

## Data Availability

The raw data supporting the conclusions of this article will be made available by the authors, without undue reservation. Mass spectrometry proteomic data were deposited to the ProteomeXchange Consortium via the PRIDE [1] partner repository with the dataset identifier PXD053543. Metabolomics data were deposited to the National Metabolomics Data Repository (https://metabolomicsworkbench.org) with the dataset identifier ST003493, the data can be accessed directly via its Project DOI: http://dx.doi.org/10.21228/M8R827.

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
