# Peer review of "A Multiomics Evaluation of the Countermeasure Influence of 4-Week Cranberry Beverage Supplementation on Exercise-Induced Changes in Innate Immunity"

_nutrients, 2024, doi:10.3390/nu16193250_

Round 1
Reviewer 1 Report
Comments and Suggestions for Authors
The article titled “A Multiomics Evaluation of the Countermeasure Influence of 4-Weeks Cranberry Beverage Supplementation on Exercise-Induced Changes in Innate Immunity” is very well written, very well planned, and also addressing a relevant topic within sports nutrition and immunology. In a broader perspective, study depicted that cranberry intake positively impacted the innate immune response and downregulated certain inflammatory markers. In contrast cranberry intake did not show promising attributes towards muscle recovery, cortisol regulation, or gut microbiome diversity, suggesting its practical application in improving exercise recovery or performance might be limited.
1- Composition of polyphenols is not provided. It is unsatisfactory while doing a clinical trial that the composition of supplement was not examined. Please analyse and provide the composition of polyphenols i.e., mg of anthocyanins, flavonols, flavonols etc. If analysis is not possible, please consult the literature.
2- How did you asses the fitness level of cyclists? If the history is not taken, then how the effects can be considered reliable?
3- Normally, Cranberry intake was considered as beneficial for gut microbiome but present study found no changes in gut microbiome? Is it a good or a negative impact? It raised a lot of questions whether a 4-week supplementation time is inadequate to generate changes in the gut microbiome, or that the polyphenol dosage was insufficient to cause significant microbiota modifications. Normally, the gut microbiota is known to be sensitive to polyphenols, especially in terms of immunological responses; this lack of impact is noticeable but not fully explained. The authors must address this questions and provide discussion in detail.
4- Sex difference was not significant? Males and females are same in number? Did authors consider hormonal factors among females? These deficiencies may have hindered the detection of meaningful differences.
5- It is always better to provide the conclusion in easier way. Please rewrite and describe the beneficial effects of cranberry polyphenols in inflammation management and also alleviating post exercise stress.
Author Response
REVIEWER #1
The article titled “A Multiomics Evaluation of the Countermeasure Influence of 4-Weeks Cranberry Beverage Supplementation on Exercise-Induced Changes in Innate Immunity” is very well written, very well planned, and also addressing a relevant topic within sports nutrition and immunology. In a broader perspective, study depicted that cranberry intake positively impacted the innate immune response and downregulated certain inflammatory markers. In contrast cranberry intake did not show promising attributes towards muscle recovery, cortisol regulation, or gut microbiome diversity, suggesting its practical application in improving exercise recovery or performance might be limited.
RESPONSE: Thank you for taking the time and effort to review our manuscript.
1- Composition of polyphenols is not provided. It is unsatisfactory while doing a clinical trial that the composition of supplement was not examined. Please analyse and provide the composition of polyphenols i.e., mg of anthocyanins, flavonols, flavonols etc. If analysis is not possible, please consult the literature.
RESPONSE: Added this statement and Supplemental Table 1:
“The cranberry and placebo beverages were supplied by the sponsor (Ocean Spray Cranberries, Inc., Lakeville-Middleboro, MA) and analyzed for polyphenol and flavonoid content. The cranberry beverage included 317±19 mg of polyphenols (measured by the Folin–Ciocalteu (F–C) colorimetric assay for total phenolic content), 294±26 mg proanthocyanidins (colorimetric assay using dimethylaminocinnamaldehyde (DMAC)), 41±5 mg anthocyanins (high-performance liquid chromatography with ultraviolet detection (HPLC-UV)), and 9 grams of intrinsic sugars per 240 ml serving. Supplemental Table 1 provides a complete polyphenol and flavonoid summary of the cranberry beverage.”
Supplemental Table 1 Polyphenol and flavonoid content of the cranberry and placebo beverages. Data are expressed as mean±SE from four batch measurements except for selected flavonoids measured once via liquid chromatography quadrupole time-of-flight.
|
Polyphenols and Flavonoids |
Cranberry Beverage (mg/240 ml) |
Placebo (mg/240 ml) |
|
*Cyanidin-3-Arabinoside |
28.1±3.5 |
ND |
|
*Cyanidin-3-Galactoside |
49.8±2.4 |
ND |
|
*Cyanidin-3-Glucoside |
<6 |
ND |
|
*Peonidin-3-Arabinoside |
24.3±2.4 |
ND |
|
*Peonidin-3-Galactoside |
67.8±10.0 |
ND |
|
*Peonidin-3-Glucoside |
5.2±1.1 |
ND |
|
*Total Anthocyanins |
41.4±4.5 |
ND |
|
**Proanthocyanidins (PACs) |
294±26.1 |
ND |
|
***Phenolic acids |
3.48 |
1.28 |
|
***Catechins |
2.04 |
1.28 |
|
***Flavonols |
33.24 |
0 |
|
****Total Polyphenols |
316.5±18.9 |
23.3±5.2 |
* High-Performance Liquid Chromatography with Ultraviolet Detection (HPLC-UV)
** Colorimetric assay using dimethylaminocinnamaldehyde (DMAC)
*** Liquid Chromatography Quadrupole Time-of-Flight (LC-QTOF)
**** Folin–Ciocalteu (F–C) assay
2- How did you assess the fitness level of cyclists? If the history is not taken, then how the effects can be considered reliable?
RESPONSE: Please see the second paragraph of the study design section 2.2. “Height and body weight were assessed, with body composition measured using the BodPod system (Cosmed, Rome, Italy). Study participants were tested for maximal aerobic capacity (VO2max) during a graded, cycling test with the Lode cycle ergometer (Lode B.V., Groningen, Netherlands) and the Cosmed CPET metabolic cart (Cosmed, Rome, Italy).”
3- Normally, Cranberry intake was considered as beneficial for gut microbiome but present study found no changes in gut microbiome? Is it a good or a negative impact? It raised a lot of questions whether a 4-week supplementation time is inadequate to generate changes in the gut microbiome, or that the polyphenol dosage was insufficient to cause significant microbiota modifications. Normally, the gut microbiota is known to be sensitive to polyphenols, especially in terms of immunological responses; this lack of impact is noticeable but not fully explained. The authors must address this questions and provide discussion in detail.
RESPONSE: Please see this paragraph (just before the conclusion paragraph) in the discussion”
“The health and anti-inflammatory effects of cranberry intake appear to depend on the gut microbiome where metabolites are generated and potential alterations in its composition occur [5]. Several randomized human clinical trials have been conducted to determine the influence of cranberry intake on gut microbiome composition and alpha and beta diversity with varying results [11,55-58]. One uncontrolled study with 28 human subjects showed that just 4 days of cranberry extract supplementation (109 mg/d polyphenols and 125 mg oligosaccharides) encouraged the growth of bifidobacteria and butyrate-producing bacteria such as Clostridium and Anaerobutyricum [9]. Further randomized controlled studies are needed. A study with mice showed that 14 days intake of a high-dose cranberry juice extract was associated with an increase of Akkermansia muciniphila and Clostridium hiranonis [56]. In general, intake of the cranberry beverage (240 ml/d) used in this 4-week study does not appear to alter the gut microbiome composition or diversity.”
Also see this statement in the conclusion section:
“Additionally, cranberry beverage intake did not alter the gut microbiome composition or diversity. Underlying mechanisms linked to these findings remain to be determined but may be associated with the dosing regimen, the macronutrient and polyphenol composition, and the presence of other bioactives within the cranberry beverage used in this study.”
In general, there are few published randomized clinical trials investigating the influence of cranberry intake on the human gut microbiome, and these do not support a consistent influence.
4- Sex difference was not significant? Males and females are same in number? Did authors consider hormonal factors among females? These deficiencies may have hindered the detection of meaningful differences.
RESPONSE: Please see the first paragraph of the results section and Table 1. “A total of 25 study participants (n=17 males, n=8 females) completed all study procedures (Table 1). Age and percent body fat were comparable between sex groups, with higher aerobic (VO2max) and ventilation capacity for the male versus female cyclists. As designed to ensure similar physiological stress states, performance data for the cranberry and placebo were comparable in average watts power, heart rates, oxy-gen consumption rates, total distance cycled, and speed (Table 2). The average percent of maximum heart rate and VO2max during both trials did not differ significantly between sex groups. Additionally, the pattern of change over time did not differ between the male and female cyclists for two key outcome measurements (total plasma oxylipins, supplement x time x sex interaction effect, p-value =0.464; gut microbiome alpha diversity, supplement x time x sex interaction effect, p-value =0.573). Thus, outcome measures are presented for all study participants combined.”
5- It is always better to provide the conclusion in easier way. Please rewrite and describe the beneficial effects of cranberry polyphenols in inflammation management and also alleviating post exercise stress.
RESPONSE: The results of this study do NOT support a beneficial effect of cranberry beverage intake on inflammation resolution following exercise stress. So, what we summarized cannot be changed. Please note that in the first paragraph of the introduction we reviewed the potential influence of cranberry polyphenols on inflammation and immunity based on cell culture studies. There is no need to repeat these statements in the conclusion paragraph.
Reviewer 2 Report
Comments and Suggestions for Authors
Thank you for submitting the manuscript "A Multiomics Evaluation of the Countermeasure Influence of 4-Weeks Cranberry Beverage Supplementation on Exercise-Induced Changes in Innate Immunity" to Nutrients.
In my opinion, the study is very interesting. It is a controlled, randomized, double-blind study that was conducted with cyclists who consumed beverages. Although the number is still small, the researchers revealed an interesting behavioral trend. I have some suggestions to clarify some doubts:
- In the abstract, it is better for the authors to provide a more comprehensive conclusion instead of including a discussion.
- The introduction could be improved by better describing the benefits of cranberries. Furthermore, I assume that there are already other studies reporting the impact of cranberries on the microbiota (although not in cyclists). Could you improve the introduction by providing a review of the subject present in the literature.
- Line#130: Was a food frequency questionnaire or 24-hour recall used?
- Line#141: Is this value an average of values ​​or was it measured only once?
- Lines144-160: Consider creating a diagram-type figure to explain in a more didactic way how the Protocol was carried out.
- Line#146: Explain better what counterbalanced is.
- Line#159: Consider including a reference to indicate the source of this recommendation.
Author Response
REVIEWER #2
Thank you for submitting the manuscript "A Multiomics Evaluation of the Countermeasure Influence of 4-Weeks Cranberry Beverage Supplementation on Exercise-Induced Changes in Innate Immunity" to Nutrients. In my opinion, the study is very interesting. It is a controlled, randomized, double-blind study that was conducted with cyclists who consumed beverages. Although the number is still small, the researchers revealed an interesting behavioral trend. I have some suggestions to clarify some doubts:
RESPONSE: Thank you for taking the time and effort to review our manuscript.
- In the abstract, it is better for the authors to provide a more comprehensive conclusion instead of including a discussion.
RESPONSE: Changed the subheading to “Conclusions.” We summarize the study in two sentences.
The introduction could be improved by better describing the benefits of cranberries. Furthermore, I assume that there are already other studies reporting the impact of cranberries on the microbiota (although not in cyclists). Could you improve the introduction by providing a review of the subject present in the literature.
RESPONSE: The entire first paragraph of the introduction reviews the potential benefits of cranberries. Regarding the gut microbiome, please note this statement in the first paragraph of the introduction:
“An increased intake of cranberry polyphenols may alter the composition of the gut microbiome, but this assumption has not yet been firmly established in randomized, placebo-controlled human clinical trials [5,9-11]. “
Also see the last paragraph of the discussion where we further review the linkage between cranberry intake and the human gut microbiome. There are few studies in this area, and generally, results indicate a null response (as we showed in our study).
- Line#130: Was a food frequency questionnaire or 24-hour recall used?
RESPONSE: Please see the second paragraph of the study design section 2.2: “A 3-d food record was completed with macro- and micro-nutrient intake assessed using the Food Processor dietary analysis software system (Version 11.11, ESHA Research, Salem, OR, USA). Urine and stool kits, and a 3-day food record were given to study participants with instructions to repeat procedures at the end of the 4-week supplementation period prior to reporting to the lab for the 2.25-h cycling bout.”
- Line#141: Is this value an average of values ​​or was it measured only once?
RESPONSE: Added this statement and Supplemental Table 1:
“The cranberry and placebo beverages were supplied by the sponsor (Ocean Spray Cranberries, Inc., Lakeville-Middleboro, MA) and analyzed for polyphenol and flavonoid content. The cranberry beverage included 317±19 mg of polyphenols (measured by the Folin–Ciocalteu (F–C) colorimetric assay for total phenolic content), 294±26 mg proanthocyanidins (colorimetric assay using dimethylaminocinnamaldehyde (DMAC)), 41±5 mg anthocyanins (high-performance liquid chromatography with ultraviolet detection (HPLC-UV)), and 9 grams of intrinsic sugars per 240 ml serving. Supplemental Table 1 provides a complete polyphenol and flavonoid summary of the cranberry beverage.”
Supplemental Table 1 Polyphenol and flavonoid content of the cranberry and placebo beverages. Data are expressed as mean±SE from four batch measurements except for selected flavonoids measured once via liquid chromatography quadrupole time-of-flight.
|
Polyphenols and Flavonoids |
Cranberry Beverage (mg/240 ml) |
Placebo (mg/240 ml) |
|
*Cyanidin-3-Arabinoside |
28.1±3.5 |
ND |
|
*Cyanidin-3-Galactoside |
49.8±2.4 |
ND |
|
*Cyanidin-3-Glucoside |
<6 |
ND |
|
*Peonidin-3-Arabinoside |
24.3±2.4 |
ND |
|
*Peonidin-3-Galactoside |
67.8±10.0 |
ND |
|
*Peonidin-3-Glucoside |
5.2±1.1 |
ND |
|
*Total Anthocyanins |
41.4±4.5 |
ND |
|
**Proanthocyanidins (PACs) |
294±26.1 |
ND |
|
***Phenolic acids |
3.48 |
1.28 |
|
***Catechins |
2.04 |
1.28 |
|
***Flavonols |
33.24 |
0 |
|
****Total Polyphenols |
316.5±18.9 |
23.3±5.2 |
* High-Performance Liquid Chromatography with Ultraviolet Detection (HPLC-UV)
** Colorimetric assay using dimethylaminocinnamaldehyde (DMAC)
*** Liquid Chromatography Quadrupole Time-of-Flight (LC-QTOF)
**** Folin–Ciocalteu (F–C) assay
- Lines144-160: Consider creating a diagram-type figure to explain in a more didactic way how the Protocol was carried out.
RESPONSE: This is a long paper with many figures and tables. We feel that our research design statement is sufficient: “This study employed a randomized, placebo controlled, double-blind, crossover design with two 4-week supplementation periods and a 2-week washout period. “ (See section 2.2).
- Line#146: Explain better what counterbalanced is.
RESPONSE: Changed to “alternative”.
- Line#159: Consider including a reference to indicate the source of this recommendation.
RESPONSE: There is no published paper on the Saris H3 direct drive smart trainer. Industry reviews claim +/- 2% power accuracy. Subjects used the same Saris trainer in this crossover study ensuring that exercise workloads were comparable between trials. We have successfully used this system in previous studies (PMID: 38873567; PMID: 38201866).